# Have Health Inequalities Increased during the COVID-19 Pandemic? Evidence from Recent Years for Older European Union Citizens

**DOI:** 10.3390/ijerph19137812

**Published:** 2022-06-25

**Authors:** Irene González Rodríguez, Marta Pascual Sáez, David Cantarero Prieto

**Affiliations:** Health Economics Research Group, Department of Economics, University of Cantabria and IDIVAL, Av. de los Castros s/n., 39005 Santander, Spain; marta.pascual@unican.es (M.P.S.); david.cantarero@unican.es (D.C.P.)

**Keywords:** health econometrics, health inequalities, self-assessed health

## Abstract

Reducing inequality is one of the current challenges that most societies are facing. Our aim was to analyze the evolution of inequalities in self-assessed health among older Europeans in a time period spanning the 2008 economic crisis and the COVID-19 health crisis. We used data from Waves 2, 4 and 8 of the Survey of Health, Ageing and Retirement in Europe. We used inequality indices that accept ordinal variables. Our empirical results suggest that average inequality declines over time. Gender significantly influences the results. Some of the countries with the highest level of inequality are Denmark and Sweden, and some with the lowest are Estonia and the Netherlands. Our results may be of interest for the development of public policies to reduce inequalities. Special attention should be paid to vulnerable groups, such as the elderly.

## 1. Introduction

### 1.1. Background

The analysis of inequality is an important field within economic science. Economists use inequality measures to answer many different questions, such as: Does the distribution of income become more inequitable over time? How do taxes affect redistribution, equity and equality? In this sense, we could differentiate studies in this area into different subcategories, such as income inequality or inequality in labor economics. In our case, we focused on the analysis of inequalities in health economics.

Among all the possible indicators, we used Self-Assessed Health (SAH), as it is one of the most widely used in the literature due to all the benefits it offers. As indicated by Fusco and Silber [1] and Pascual et al. [2], it is a very accessible variable, as most surveys include it in their questionnaires. On the other hand, it allows covering a large part of the individual’s level of health. Finally, it is a good predictor of morbidity and mortality [1,2]. The challenge with this variable is that it is a categorical variable. This makes some traditional indicators of inequality measurement, such as the concentration index, unusable, as they accept only continuous variables [3]. Several authors, such as Makdissi and Yazbeck [4] and Madden [5], discussed why some indices do not accept categorical variables and proposed solutions: an adaptation of the indices or new indices compatible with non-continuous variables.

The aim of this study was to analyze trends in health inequality in recent years in European Union countries, comparing the pre-pandemic and the pandemic situation. For this purpose, we provided an empirical illustration of health inequality using data from the Survey on Health, Ageing and Retirement in Europe in three different waves: the second (2006), the sixth (2010) and the eighth (2020). In addition, the samples were subdivided according to gender to understand differences between groups. Therefore, this paper contributes to the use of inequality indicators in the field of health economics and to the analysis of their evolution in recent years, which may be of interest for public policy-making.

The paper is, therefore, structured as follows. Section 1.2 reviews the existing literature about health inequality, where the main research areas of interest are identified, while Section 1.3 describes the data and defines the key variable: self-assessed health. Next, we develop the measurement of inequality used. The manuscript concludes with the main results of the analysis and discussion in Section 3 and Section 4. Finally, Section 5 presents some concluding remarks.

### 1.2. Previous Literature

To describe the existing literature on this issue and as our sample was European countries, all reviewed studies also partially or fully analyzed these countries.

First, we found some studies focused on health inequalities. Many years before the economic crisis, Doorslaer and Koolman [6] analyzed the differences in income-related health inequalities across European countries with data taken from the 1996 wave of the European Community Household Panel. The results suggest that significant health inequalities favoring higher-income groups exist in all countries. In particular, they are high in Portugal, the United Kingdom and Denmark, while they are relatively low in the Netherlands and Germany. Although a positive correlation was found with income inequality, the authors pointed out that health inequality does not come only from income.

Šućur and Zrinščak [7] compared differences in access to health care and self-reported health status between Croatia and other European Union countries before the 2008 economic crisis. They compared statistical differences between categories (income groups, urban–rural divide and analytical regions in the case of Croatia) and found significant results for all the variables under study. As a result, they warned that results are worse in Croatia than in the European comparator countries. As an example, the authors indicated that the *“rural urban proportion ratio of those who reported poor health was about 80% higher in Croatia than in both European Union country groups”.*

On the other hand, convergence methods have also been used for the analysis of health inequalities. This is the case of Jaworska [8], who evaluated the β-convergence of life expectancy between European regions in the period 2002–2012. She found that there was such a process and that the regions with the lowest life expectancy values initially grew the most. However, the process was not the same for all regions, which could influence the level of existing inequality.

Motivated by the unfavorable health trends of the low-educated population in the United States, Mackenbach et al. [9] studied the relationship between health trends and the educational level of Europeans. The period analyzed (1980–2014) included the economic crisis of 2008, which allowed the authors to analyze the impact of crisis-related economic conditions on health outcomes. Their results suggest that Europe did not repeat the same trend as the United States. They also pointed out that there did not seem to be any short-term effects of the crisis on population health in most European countries.

In a shorter period, but also covering the years after the 2008 economic crisis, Pascual et al. [2] provided an empirical illustration of health inequality and polarization using data from 27 countries from the European Health Interview Survey. The authors analyzed self-assessed health and, as its categorical variable, they used median-based indices. The results indicate that, among the countries with “very good” health, Greece had the highest level of polarization and health inequality. In addition, inequalities were found to increase in some European countries, such as Spain, Romania and Ireland, between 2006–2009 and 2013–2015.

In addition, Pinillos-Franco and Somarriba-Arechavala [10] created their own health indicator based on the P_2_ distance method. In this way, they compared the results covering different dimensions between European countries (and between genders) to capture possible existing health inequalities. They found that northern and southern countries have the highest levels of health. Inequalities might follow a territorial pattern.

As a review of the literature has shown, not many studies apply inequality indices to analyses of health, and even fewer focus on the European population. Other methodologies, such as convergence methods or traditional regressions, have generally been applied to health inequality analyses. Therefore, to contribute to the use of inequality indices in health inequality analyses, we applied six different measures for the European adult population. Table 1 presents the main findings of the reviewed studies.

### 1.3. Data

To study health inequality, we used data from the Survey of Health, Ageing and Retirement in Europe (SHARE), a database enabling analysis of the effects of health, social, economic and environmental policies across the life course of European senior citizens. The questionnaire has an interdisciplinary character, as it covers many variables that can be arranged in the following categories: health, biomarkers, psychological variables, economic variables and social support variables. To date, it is the largest pan-European social science survey providing internationally longitudinal micro-data. Specifically, from 2004 to the present, 530,000 interviews have been conducted with 140,000 people aged 50 and over in 28 European countries and Israel.

To capture the evolution of health in the period of the 2008 economic crisis and the COVID-19 health crisis, Wave 2 (2006), Wave 4 (2010) and Wave 8 (2020) were selected (Appendix A). As indicated above and for all the advantages it offers, among all the variables, we made use of self-perceived health. This indicator is divided into five categories according to the state of health: it takes the value 1 if the individual considers his or her level of health to be excellent, 2 if it is very good, 3 if good, 4 if fair and, finally, 5 if poor. In Table 2, Table 3 and Table 4, we can see the distribution of self-perceived health by year, country and gender. The sample size of Wave 2 is 33,099, Wave 4 is 54,046 and Wave 8 is 45,281.

## 2. Methodology

Following the economic literature, there are many indicators of inequality. Although the traditional ones are those proposed by Theil [11] and Foster, Greer and Thorbecke [12], the variable under analysis in this study was categorical, and therefore not all indices could be used without presenting various problems. In this sense, the following indicators were calculated. In all of them, we assumed that *p_i_* is the cumulative proportion of individuals of the sample in each category *i*, where *i* = 1, …, *m* (in our case *m* = 5) and *m_e_* is the median.

The first of these is the index developed in Apouey [13] and Abul Naga and Yalcin [14], which can be defined such that:(1)$$1-\frac{2\sum_{i=1}^{m}\left|p_{i}-0.5\right|-1}{m-1}\;\;\;\;\;\;\;\;\;$$

The following three indicators were developed in Reardon [15]:(2)$$-\frac{\sum_{i=1}^{m-1}\left(p_{i}\;log_{2}\;p_{i}+\left(1-p_{i}\right)\;log_{2}\left(1-p_{i}\right)\right)}{m-1}\;\;\;\;\;\;\;\;\;$$
(3)$$\frac{\sum_{i=1}^{m-1}4\;p_{i}\;\left(1-p_{i}\right)}{m-1}\;\;\;\;\;\;\;\;\;$$
(4)$$\frac{2\sum_{i=1}^{m-1}\sqrt{p_{i}\;\left(1-p_{i}\right)}}{m-1}\;\;\;$$

The fifth index was developed by Abul Naga and Yalcin [14]:(5)$$\frac{\sum_{i<m_{e}}p_{i}^{\alpha }-\sum_{i\geq m_{e}}p_{i}^{\beta }+\left(m+1-m_{e}\right)}{k_{\alpha ,\beta }+\left(m+1-m_{e}\right)}\;,\;\;\alpha ,\beta \geq 1\;$$
where
(6)$$k_{\alpha ,\beta }=\left(m_{e}-1\right){\left(\frac{1}{2}\right)}^{\alpha }-\left[1+\left(m-m_{e}\right){\left(\frac{1}{2}\right)}^{\beta }\right]$$

The above equality is a normalization which ensures that the index values lie between 0 and 1. The authors noted that the two parameters, *α* and *β*, are included to allow the researcher to accommodate differing judgments regarding inequality above and below the median. In this sense, less weight will be given to inequalities above the median for higher values of both parameters. In our analysis, we present two different situations. In the first one, the parameters take the same value (equal to 1); therefore, we consider that the index is symmetric (equal deviations from 0.5 below and above the median are judged as being equivalent in terms of inequality). In the second scenario, we give more weight to inequalities below the median to give more importance to the lower levels following health economics literature such as Wagstaff [16]; therefore, *α* takes the value 1 and *β* takes the value 4.

Finally, the last index used in this analysis is the one proposed by Lazar and Silber [17]:(7)$$\;\;\frac{\sum_{i<m_{e}}{\left(2p_{i}\right)}^{\alpha }+\sum_{i\geq m_{e}}{\left(2\left(1-p_{i}\right)\right)}^{\beta }}{m-1}\;\;\;\;$$

With the latter index, two different scenarios are also proposed. In the first scenario, both *α* and *β* take the value 1.5. In the second, *α* remains 1.5, but *β* takes the value 2, as proposed by Lazar and Silber (2013) in their study.

It should be noted that, as analyzed by Wang and Xu [18], all the indices satisfy normalization, invariance to parallel shifts and simple aversion to median-preserving spreads. Additionally, indicators 1 (I1) and 3 (I3) satisfy additivity, and only 3 (I3) also satisfies independence.

## 3. Results

In this section, we provide an empirical illustration of the use of the indices developed above. Table 5, Table 6, Table 7, Table 8, Table 9 and Table 10 show the index results for the European countries for which data are available for the second wave (2006), fourth wave (2010) and eighth wave (2020). The first column refers to the country, the second to the median value (denoted as *m*) and the following to the six indicators. It should be noted that, as indicated above, for the last two indicators (5 and 6), two different situations are analyzed depending on the value of the parameters.

Regarding Wave 2 data (year 2006), the three countries with the highest average values for inequality indicators are Denmark, Ireland and Sweden (for both men and women). On the other hand, the three countries with the lowest inequality for men are Germany, Greece and Belgium. This ranking is different for women, which can be ordered as Belgium, the Netherlands and Spain. This indicates that there are no gender differences among the most unequal countries, but there are differences among the least unequal countries. If we compare across indices, we see that they do not all have the same sensitivity, nor do they rank countries in the same way. It is true that all indices rank Denmark, Ireland or Sweden in some cases as the most unequal country (for both genders). However, the position for other countries is not so clear. For men, Poland and Spain are the two countries with the greatest variability. In some cases, they rank fifth, while in others, they rank last (position 13). For women, the same is true for Poland.

Secondly, we see that in 2010 (Wave 4), the ranking of countries according to SAH inequality changed. The three most unequal countries for men are Denmark, Sweden and Italy, while for women, they are Sweden, Denmark and Slovenia. Otherwise, the three least unequal countries are Poland, Portugal and Estonia for men and the Netherlands, Portugal and Estonia for women. Each indicator also offers its own ranking. All of them put Estonia in the last position, i.e., they rank it as the least unequal country, for both genders. In the case of men, the country with the largest change in rank according to the index used is Belgium (from seventh to thirteenth position), and in the case of women, it is the Czech Republic and Hungary (from fifth to eleventh position).

Finally, we can observe the results for the last wave published by the SHARE, which correspond to information collected in 2020. The three most unequal countries on average are Denmark, Cyprus and Croatia, while the three least unequal are Lithuania, Estonia and Latvia (the rankings are the same for both genders). As in the previous cases, there are countries which all six indicators rank in a similar position, while there are others that have a greater variation. For men, Slovakia and Poland are sometimes ranked 11th and sometimes 21st. For women, Poland is the country with the greatest variation, ranking 8th and 21st.

If we compare the average of all the countries analyzed for each indicator, we see that in all cases, the inequality values are decreasing. The growth rates from 2006 to 2010 and from 2010 to 2020 are in all cases negative, except for indicators I1 and I5-1 for women, where they show no change (zero growth rate). This means that, despite the existence of inequalities in SAH, the trend in recent years is decreasing. The biggest drop for men is I6-2 between 2006 and 2010 (−10%) and for women between 2010 and 2020 for indicators I1 and I5-1 (also a growth rate of −10%). Figure 1 and Figure 2 show the averages for each indicator for all countries. Figure 3, Figure 4 and Figure 5 show the ranking position of each country for each wave.

## 4. Discussion

Having analyzed the evolution of inequalities in SAH, our findings suggest that there are inequalities in SAH among European adult citizens both before the 2008 economic crisis and during the COVID-19 health crisis. This relevant result is related to what has been found in the literature [2]. However, by using such recent data (2020), our study takes a current view of the situation, and we provide an important new result: the growth rates of inequalities in SAH are not positive in the 2006–2020 period. A few years ago, Hu et al. [19] analyzed trends in socioeconomic inequalities in self-assessed health in 17 European countries, and their results suggest that “a better understanding of the causes of these inequalities is needed in order to develop policies or interventions that effectively reduce inequalities in SAH”. In this sense, our study suggests that policy makers may have been better able to identify the causes of these inequalities and, through corresponding public policies, are succeeding in reducing them over the years.

Results vary depending on the selected indicator, gender and time period analyzed. If we focus on gender differences, in Wave 2 (2006), in 19.71% of the indicators and countries studied, women have higher values of inequality, while in 15.87% of the analyses performed, men have higher values. This gap increased significantly after the 2008 crisis. In 2010, in 37.92% of the analyses, women have a higher rate of inequality compared to 6.25% of the analyses of men. However, in 2020, at the height of the COVID-19 health crisis, this trend is reversed. Men have a higher value in 28% of the analyses, while this value is 12.57% for women. Our insights are in line with the conclusions made by other authors in other European countries, such as Crimmins et al. [20]. They claimed that there are gender differences in many variables that characterize the health of individuals, such as SAH.

Another relevant result concerns geographical differences, as was already found in the existing literature [10,21,22]. Not all countries show the same trend or values, but there are different results depending on the country analyzed. Those that tend to have the highest values of inequality are Denmark and Sweden, while some with lower inequality values are Estonia and the Netherlands. These differences may be explained by different socio-economic characteristics of the population or by different existing public policies. As other authors point out [23], multidisciplinary collaborations in creating public policies are essential.

Some limitations should also be mentioned. First of all, we point out that the results vary depending on the indicator selected. They should be interpreted as a whole and compared with each other to try to reduce bias as much as possible. On the other hand, each wave of the SHARE does not cover the same countries, and therefore our analyses are not always on the same sample.

## 5. Conclusions

The main objective of this study was to analyze the evolution of inequality in health. For this purpose, we used data from Waves 2, 4 and 8 of the SHARE (for 2006, 2010 and 2020, respectively) for different European countries. As our variable under study was an ordinal variable (SAH), we had to select inequality measurement indicators that can be applied in this type of situation. Specifically, we used those developed by Apouey [13], Abul Naga and Yalcin [14], Reardon [15] and Lazar and Silber [17]. Furthermore, we did not only analyze six different indicators, but for two of them, we also assumed two different situations by giving different values to the α and β parameters. With all this, we could construct a ranking from the most unequal to the least unequal country and examine the sensitivity of the results according to the indicator used.

Our results suggest that inequalities in SAH have existed and still do among older European citizens. We can state that the results are sensitive to the indicator used, the time period and gender. Just before the economic crisis of 2008, in 2006, the most unequal countries on average were Denmark, Ireland and Sweden for both men and women. After the crisis, in 2010, the three countries with the highest inequality on average for men are Denmark, Sweden and Italy and for women Sweden, Denmark and Slovenia. Finally, during the COVID-19 health crisis, the three countries with the highest inequality in SAH on average are Denmark, Cyprus and Croatia for both genders. This shows that the most unequal country, on average, by a large majority is Denmark (in all but one case it ranks first). However, the three countries with the lowest inequality have much greater variability by year studied.

Referring to gender differences, in 2006 and 2010, inequality on average was higher for women than for men. In 2020, this trend is reversed, and inequality is higher for men than for women. On the other hand, comparing the growth rates from one time period to another, we see that none of them is positive, which suggests that, although inequalities in SAH still exist, the evolution is favorable because inequality values are decreasing.

Furthermore, as possible future research, socio-economic, cultural or environmental factors that positively and negatively affect individuals’ SAH could be identified and measures of inequality decomposed. The effects of public policies on health inequalities could also be assessed. In this regard, it is worth noting that in the wake of the pandemic, major international organizations are discussing how public health and social determinants could be improved across Europe. This is the case of EuroHealthNet [24], which is looking at how countries could come together to create enabling environments for wellbeing, with equity and wellbeing as key pillars for the development and promotion of such public policies. The analysis could also be replicated for other variables characterizing population health. A special effort should be made to ensure an adequate level of health for vulnerable people, such as the elderly.

## Figures and Tables

**Figure 1 ijerph-19-07812-f001:**
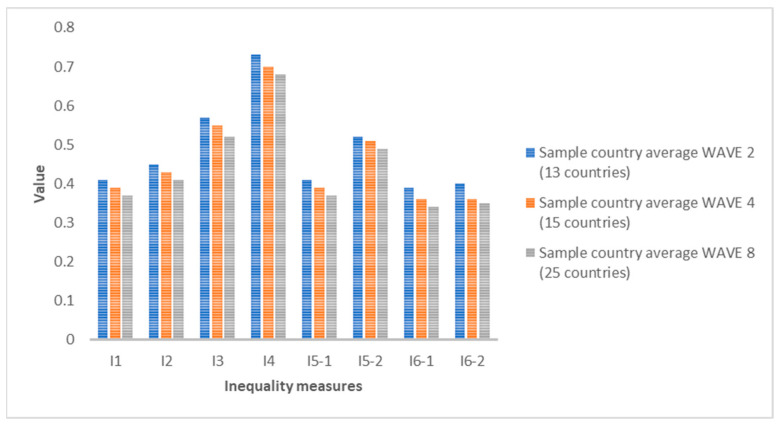
Average values for each inequality indicator for men in Wave 2 (2006), Wave 4 (2010) and Wave 8 (2020). Source: Authors’ elaboration based on SHARE (2021).

**Figure 2 ijerph-19-07812-f002:**
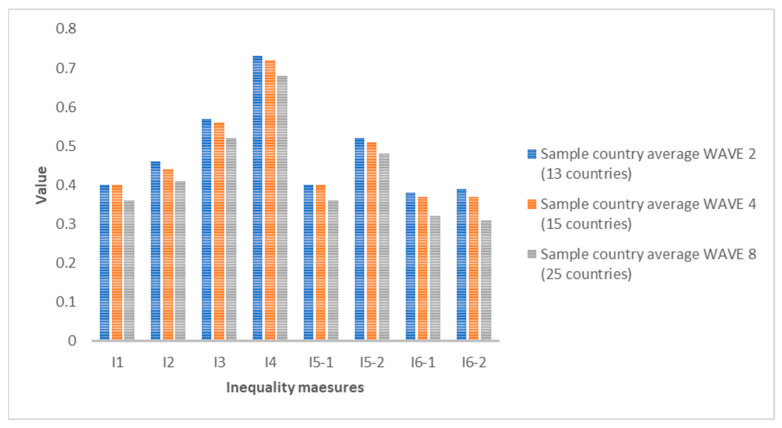
Average values for each inequality indicator for women in Wave 2 (2006), Wave 4 (2010) and Wave 8 (2020). Source: Authors’ elaboration based on SHARE (2021).

**Figure 3 ijerph-19-07812-f003:**
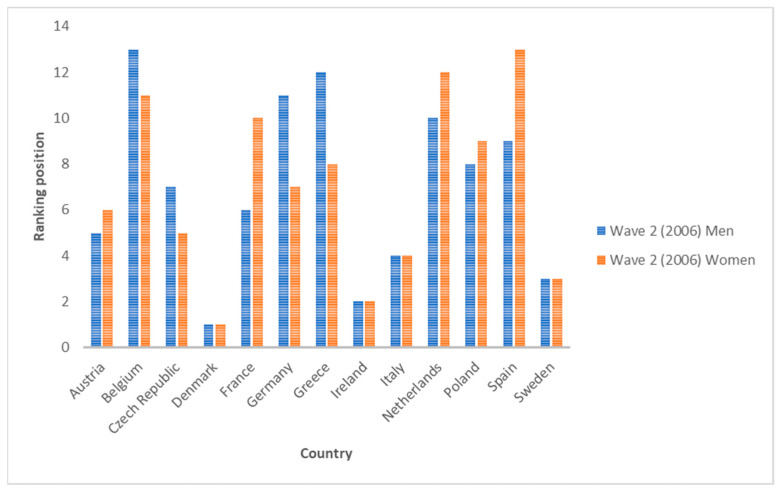
Position in the ranking of each country in Wave 2 (2006). Source: Authors’ elaboration based on SHARE (2021).

**Figure 4 ijerph-19-07812-f004:**
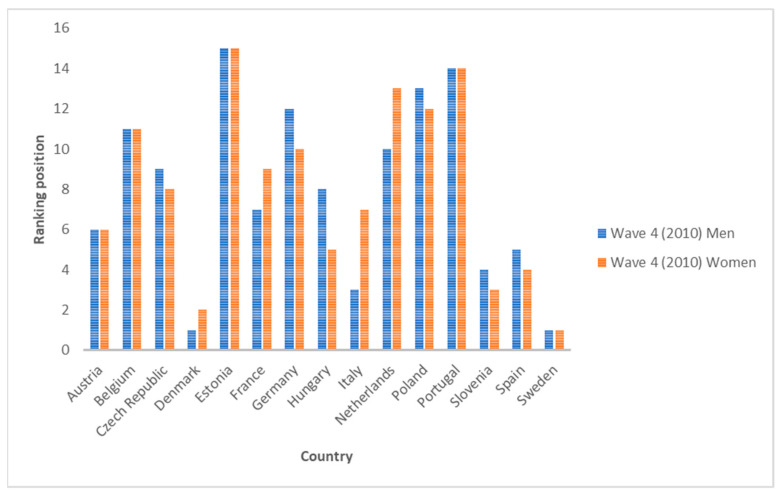
Position in the ranking of each country in Wave 4 (2010). Source: Authors’ elaboration based on SHARE (2021).

**Figure 5 ijerph-19-07812-f005:**
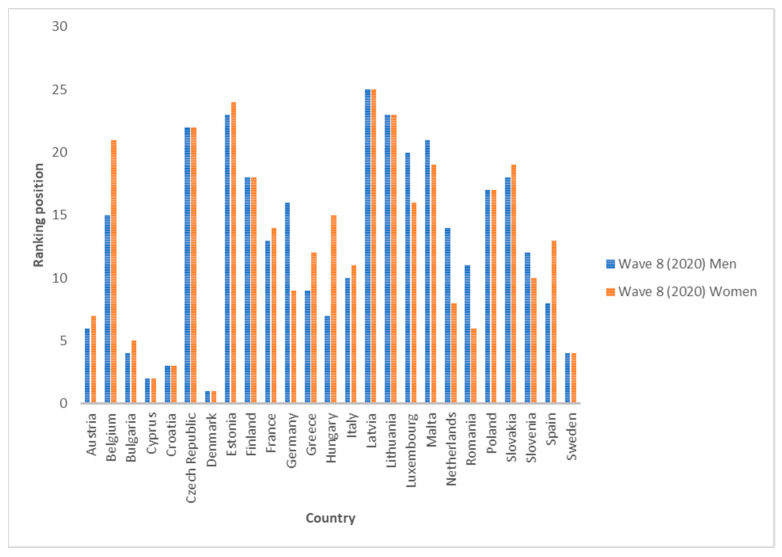
Position in the ranking of each country in Wave 8 (2020). Source: Authors’ elaboration based on SHARE (2021).

**Table 1 ijerph-19-07812-t001:** Main conclusions of the studies about health inequality review.

Authors	Year	Country	Methodology	Main Results
Doorslaer and Koolman [6]	1996	13 European countries	Health concentration index	Significant inequalities in health favoring the higher-income groups emerge in all countries.
Šućur and Zrinščak [7]	2003 and 2006	European countries	χ^2^ test or analysis of variance	Health inequalities are much more pronounced in Croatia than in European Union countries. Significant differences were found among 4 income quartiles in all health indicators under study.
Jaworska [8]	2002–2012	28 European countries	β-convergence analysis	A β-convergence process of life expectancy has taken place in the European Union regions.
Mackenbach et al. [9]	1980–2014 2002–2014	European countries	Interrupted time-series and country-fixed effects analyses	There has not been short-term impact of the 2008 economic crisis on health inequalities at the population level.
Pascual et al. [2]	2006–2009 2013–2015	27 European countries	Abul Naga and Yalcin (2008) inequality index and a polarization index proposed by Apouey (2010) and Apouey and Silber (2013)	Inequality is increasing in countries such as Greece, Ireland, Romania and Spain, among others.
Pinillos-Franco and Somarriba-Arechavala [10]	2012	28 European countries	Their own indicator based on the P2 distance method	There is a territorial pattern across Europe (eastern countries have poorer levels of health).

Source: Authors’ elaboration.

**Table 2 ijerph-19-07812-t002:** Distribution of self-perceived health by gender and country in Wave 2 (2006).

Country	Wave 2 (2006)									
	Men					Women				
	Excellent	Very Good	Good	Fair	Poor	Excellent	Very Good	Good	Fair	Poor
Austria	7.01	19.50	39.27	25.39	8.84	8.07	21.12	38.72	23.81	8.28
Belgium	7.71	19.49	42.32	23.57	6.91	9.99	19.85	42.85	20.88	6.43
Czech Republic	3.47	14.96	38.95	30.37	12.25	2.71	17.29	37.90	27.07	15.02
Denmark	19.29	31.54	23.79	19.29	6.09	20.61	32.15	24.94	16.03	6.28
France	7.16	13.44	43.47	25.52	10.42	8.72	13.48	44.01	24.19	9.60
Germany	5.49	15.99	40.99	27.96	9.58	5.83	16.25	41.25	25.83	10.83
Greece	6.49	26.52	39.24	21.94	5.82	9.45	31.92	36.03	17.40	5.21
Ireland	22.12	29.56	26.02	15.93	6.37	20.86	30.32	28.82	15.27	4.73
Italy	6.12	12.2	33.33	34.19	14.13	8.63	13.62	38.69	27.53	11.53
The Netherlands	12.39	15.47	42.64	24.64	4.86	12.09	15.05	43.83	24.18	4.85
Poland	1.08	6.34	29.90	28.46	34.22	1.40	6.55	30.24	29.68	23.12
Spain	2.61	9.16	37.08	33.43	17.72	3.34	12.63	42.53	29.71	11.79
Sweden	16.75	20.78	30.08	24.60	7.78	18.34	24.61	31.74	19.12	6.19

Source: Authors’ elaboration based on SHARE (2021).

**Table 3 ijerph-19-07812-t003:** Distribution of self-perceived health by gender and country in Wave 4 (2010).

Country	Wave 4 (2010)									
	Men					Women				
	Excellent	Very Good	Good	Fair	Poor	Excellent	Very Good	Good	Fair	Poor
Austria	9.99	24.78	35.40	24.08	5.76	8.46	27.22	33.49	23.33	7.50
Belgium	6.57	19.21	41.72	24.93	7.56	8.59	22.40	41.41	21.71	6.44
Czech Republic	2.74	14.01	38.99	30.24	14.01	3.20	15.57	37.76	28.51	14.96
Denmark	19.60	35.08	22.82	17.42	5.08	20.98	33.11	25.22	15.01	5.68
Estonia	1.32	4.24	23.05	49.60	21.80	1.63	3.63	24.18	48.62	21.93
France	6.47	14.05	41.60	25.74	12.15	7.27	15.18	41.82	24.16	11.57
Germany	3.94	11.48	43.16	32.02	9.40	3.84	13.89	38.89	32.41	10.98
Hungary	3.37	8.57	25.20	37.77	25.09	3.73	10.95	27.98	34.52	22.81
Italy	5.60	15.52	34.66	30.43	13.79	9.01	14.77	39.11	27.35	9.76
The Netherlands	11.49	15.34	43.07	24.84	5.26	12.08	18.29	41.63	24.49	3.51
Poland	0.41	6.78	34.50	35.01	23.31	1.19	7.16	35.28	33.95	22.41
Portugal	2.19	6.31	27.17	42.42	21.91	3.87	8.79	29.31	43.26	14.77
Slovenia	4.91	12.78	38.02	29.57	14.72	7.35	13.02	37.70	26.71	15.22
Spain	3.02	12.32	33.46	32.73	18.46	4.46	15.65	37.51	28.12	14.27
Sweden	14.68	23.14	27.56	26.43	8.18	16.35	25.70	28.36	20.13	9.45

Source: Authors’ elaboration based on SHARE (2021).

**Table 4 ijerph-19-07812-t004:** Distribution of self-perceived health by gender and country in Wave 8 (2020).

Country	Wave 8 (2020)									
	Men					Women				
	Excellent	Very Good	Good	Fair	Poor	Excellent	Very Good	Good	Fair	Poor
Austria	5.95	22.85	34.86	26.89	9.46	7.13	20.75	38.74	24.31	9.08
Belgium	5.52	29.60	41.55	26.38	6.96	6.00	20.81	45.48	22.96	4.75
Bulgaria	2.80	16.64	35.89	27.29	17.38	3.97	21.25	36.54	25.78	12.46
Croatia	2.71	13.83	34.44	29.62	19.40	4.70	14.48	33.66	33.27	13.89
Cyprus	5.31	16.56	30.63	34.69	12.81	10.29	19.12	34.31	25.00	11.27
Czech Republic	2.73	15.27	53.64	22.30	6.06	1.81	14.71	53.77	20.73	8.98
Denmark	17.16	33.39	24.51	19.47	5.47	17.11	32.09	27.53	17.00	6.28
Estonia	1.57	4.13	23.90	51.57	18.83	1.46	2.74	21.50	51.33	22.96
Finland	7.48	13.82	41.79	32.36	4.55	5.05	11.40	41.68	35.33	6.54
France	6.36	14.66	41.56	28.42	8.99	5.85	13.95	45.56	24.88	9.76
Germany	4.11	15.39	40.70	31.77	8.02	3.89	15.25	38.79	31.69	10.39
Greece	5.09	23.87	36.81	26.33	7.90	5.72	24.94	38.36	23.35	7.63
Hungary	2.36	11.99	38.33	31.26	16.06	2.65	12.58	43.38	31.13	10.26
Italy	3.64	14.49	33.03	38.91	9.93	5.44	12.90	39.02	33.80	8.85
Latvia	0.21	1.45	20.29	50.93	27.12	0.00	1.74	27.53	52.61	18.12
Lithuania	0.90	4.38	36.14	46.69	11.90	1.91	4.97	34.80	48.37	9.94
Luxembourg	3.85	18.50	42.39	27.55	7.71	6.07	17.76	42.29	27.10	6.78
Malta	1.62	16.67	36.34	41.44	3.94	2.81	21.35	37.36	34.55	3.93
The Netherlands	11.97	16.09	42.53	24.52	4.89	13.33	16.09	41.84	22.99	5.75
Poland	0.70	6.99	41.17	34.62	16.52	0.44	6.09	41.53	34.33	17.61
Romania	1.83	6.60	39.47	28.23	23.88	2.06	13.08	42.06	23.74	19.07
Slovakia	8.52	20.19	48.33	15.56	7.41	10.02	20.27	49.43	15.49	4.78
Slovenia	4.11	15.21	42.47	26.23	11.99	4.42	14.55	43.85	24.39	12.78
Spain	2.03	11.90	37.55	33.25	15.27	4.13	14.73	40.63	30.80	9.71
Sweden	14.04	23.03	33.60	22.95	6.39	12.85	24.21	35.58	21.81	5.55

Source: Authors’ elaboration based on SHARE (2021).

**Table 5 ijerph-19-07812-t005:** Inequality indicators in self-perceived health in Wave 2 (2006) for men.

Country	Men											
	m	I1	I2	I3	I4	I5-1	I5-2	I6-1	I6-2	Variance	Standard Error	Mean
**Poland**	4	0.40 (5)	0.39 (13)	0.54 (9)	0.66 (13)	0.40 (5)	0.52 (6)	0.36 (6)	0.34 (7)	0.01	0.11	0.45
**Spain**	4	0.40 (5)	0.40 (12)	0.52 (13)	0.68 (12)	0.40 (5)	0.48 (10)	0.41 (5)	0.39 (5)	0.01	0.10	0.46
**Austria**	3	0.38 (7)	0.45 (4)	0.57 (5)	0.73 (5)	0.38 (7)	0.51 (7)	0.34 (8)	0.32 (8)	0.02	0.14	0.46
**Belgium**	3	0.36 (13)	0.43 (9)	0.55 (7)	0.71 (8)	0.36 (13)	0.47 (11)	0.31 (12)	0.29 (12)	0.02	0.14	0.44
**Czech Republic**	3	0.38 (7)	0.42 (11)	0.54 (9)	0.70 (11)	0.38 (7)	0.53 (5)	0.36 (6)	0.37 (6)	0.01	0.12	0.46
**France**	3	0.37 (9)	0.44 (5)	0.55 (7)	0.72 (6)	0.37 (9)	0.51 (7)	0.33 (9)	0.32 (8)	0.02	0.14	0.45
**Germany**	3	0.37 (9)	0.43 (9)	0.54 (9)	0.71 (8)	0.37 (9)	0.50 (9)	0.33 (9)	0.32 (8)	0.02	0.13	0.45
**Greece**	3	0.37 (9)	0.44 (5)	0.54 (9)	0.70 (10)	0.37 (9)	0.46 (12)	0.31 (12)	0.29 (12)	0.02	0.13	0.44
**Italy**	3	0.43 (4)	0.44 (5)	0.58 (4)	0.74 (4)	0.43 (4)	0.57 (3)	0.43 (4)	0.48 (3)	0.01	0.11	0.51
**The Netherlands**	3	0.37 (9)	0.44 (5)	0.56 (6)	0.72 (6)	0.37 (9)	0.46 (12)	0.32 (11)	0.30 (11)	0.02	0.14	0.44
**Sweden**	3	0.47 (3)	0.52 (3)	0.66 (1)	0.80 (1)	0.47 (3)	0.56 (4)	0.46 (3)	0.44 (4)	0.02	0.12	0.55
**Denmark**	2	0.50 (1)	0.54 (2)	0.65 (2)	0.78 (3)	0.50 (1)	0.62 (1)	0.57 (1)	0.71 (1)	0.01	0.10	0.61
**Ireland**	2	0.50 (1)	0.55 (1)	0.65 (2)	0.79 (2)	0.50 (1)	0.61 (2)	0.55 (2)	0.67 (2)	0.01	0.10	0.60
**Sample country average (13 countries)**	3	0.41	0.45	0.57	0.73	0.41	0.52	0.39	0.40	-	-	-

The inequality rank appears in parentheses (the most unequal country takes the value 1). Source: Authors’ elaboration based on SHARE (2021).

**Table 6 ijerph-19-07812-t006:** Inequality indicators in self-perceived health in Wave 2 (2006) for women.

Country	Women											
	m	I1	I2	I3	I4	I5-1	I5-2	I6-1	I6-2	Variance	Standard Error	Mean
**Poland**	4	0.40 (6)	0.39 (13)	0.54 (12)	0.67 (13)	0.40 (5)	0.52 (6)	0.36 (6)	0.34 (6)	0.01	0.11	0.45
**Spain**	3	0.36 (13)	0.41 (12)	0.51 (13)	0.68 (12)	0.36 (13)	0.51 (8)	0.34 (8)	0.34 (6)	0.01	0.12	0.44
**Austria**	3	0.39 (7)	0.46 (4)	0.57 (5)	0.73 (5)	0.39 (7)	0.51 (8)	0.34 (8)	0.32 (10)	0.02	0.14	0.46
**Belgium**	3	0.37 (10)	0.44 (7)	0.56 (6)	0.72 (7)	0.37 (10)	0.47 (11)	0.32 (11)	0.29 (12)	0.02	0.14	0.44
**Czech Republic**	3	0.40 (5)	0.44 (7)	0.56 (6)	0.71 (10)	0.40 (5)	0.55 (3)	0.38 (4)	0.40 (4)	0.01	0.12	0.48
**France**	3	0.37 (10)	0.44 (7)	0.56 (6)	0.73 (5)	0.37 (10)	0.50 (10)	0.32 (11)	0.31 (11)	0.02	0.14	0.45
**Germany**	3	0.38 (9)	0.44 (7)	0.56 (6)	0.72 (7)	0.38 (9)	0.52 (6)	0.34 (8)	0.33 (8)	0.02	0.13	0.46
**Greece**	3	0.39 (8)	0.46 (4)	0.55 (11)	0.71 (10)	0.39 (7)	0.47 (11)	0.36 (6)	0.33 (8)	0.02	0.12	0.46
**Italy**	3	0.41 (4)	0.46 (4)	0.59 (4)	0.75 (4)	0.41 (4)	0.54 (4)	0.38 (4)	0.38 (5)	0.02	0.13	0.49
**The Netherlands**	3	0.37 (10)	0.43 (11)	0.56 (6)	0.72 (7)	0.37 (10)	0.46 (13)	0.31 (13)	0.29 (12)	0.02	0.14	0.44
**Sweden**	3	0.46 (3)	0.52 (2)	0.64 (2)	0.78 (1)	0.46 (3)	0.53 (5)	0.46 (3)	0.44 (3)	0.01	0.12	0.54
**Denmark**	2	0.48 (1)	0.54 (1)	0.65 (1)	0.78 (1)	0.48 (1)	0.60 (1)	0.53 (1)	0.64 (1)	0.01	0.10	0.59
**Ireland**	2	0.47 (2)	0.52 (2)	0.62 (3)	0.76 (3)	0.47 (2)	0.58 (2)	0.51 (2)	0.61 (2)	0.01	0.10	0.59
**Sample country average (13 countries)**	3	0.40	0.46	0.57	0.73	0.40	0.52	0.38	0.39	-	-	-

The inequality rank appears in parentheses (the most unequal country takes the value 1). Source: Authors’ elaboration based on SHARE (2021).

**Table 7 ijerph-19-07812-t007:** Inequality indicators in self-perceived health in Wave 4 (2010) for men.

Country	Men											
	m	I1	I2	I3	I4	I5-1	I5-2	I6-1	I6-2	Variance	Standard Error	Mean
**Estonia**	4	0.29 (15)	0.33 (15)	0.44 (15)	0.60 (15)	0.29 (15)	0.40 (15)	0.22 (15)	0.20 (15)	0.02	0.13	0.35
**Hungary**	4	0.39 (7)	0.40 (11)	0.56 (6)	0.71 (8)	0.39 (7)	0.50 (7)	0.36 (7)	0.33 (9)	0.02	0.13	0.46
**Poland**	4	0.36 (11)	0.38 (13)	0.49 (14)	0.62 (14)	0.36 (11)	0.47 (12)	0.32 (10)	0.30 (11)	0.01	0.11	0.41
**Portugal**	4	0.34 (14)	0.37 (14)	0.50 (12)	0.66 (13)	0.34 (14)	0.45 (14)	0.30 (14)	0.27 (14)	0.02	0.13	0.40
**Spain**	4	0.43 (3)	0.43 (7)	0.56 (6)	0.71 (8)	0.43 (3)	0.50 (7)	0.45 (3)	0.42 (4)	0.01	0.10	0.49
**Austria**	3	0.40 (6)	0.46 (3)	0.58 (3)	0.73 (4)	0.40 (6)	0.49 (9)	0.36 (7)	0.34 (8)	0.02	0.13	0.47
**Belgium**	3	0.36 (11)	0.43 (7)	0.54 (10)	0.71 (8)	0.36 (11)	0.48 (11)	0.31 (13)	0.29 (13)	0.02	0.14	0.44
**Czech Republic**	3	0.39 (8)	0.42 (10)	0.53 (11)	0.69 (11)	0.39 (7)	0.54 (5)	0.38 (6)	0.40 (6)	0.01	0.11	0.47
**France**	3	0.39 (8)	0.44 (5)	0.57 (5)	0.73 (4)	0.39 (7)	0.53 (6)	0.35 (9)	0.35 (7)	0.02	0.13	0.47
**Germany**	3	0.35 (13)	0.39 (12)	0.50 (12)	0.67 (12)	0.35 (13)	0.49 (9)	0.32 (10)	0.32 (10)	0.01	0.12	0.42
**Italy**	3	0.42 (4)	0.45 (4)	0.58 (3)	0.74 (3)	0.42 (4)	0.56 (3)	0.41 (4)	0.43 (3)	0.01	0.12	0.50
**The Netherlands**	3	0.37 (10)	0.43 (7)	0.56 (6)	0.72 (6)	0.37 (10)	0.47 (12)	0.32 (10)	0.29 (12)	0.02	0.14	0.44
**Slovenia**	3	0.41 (5)	0.44 (5)	0.56 (6)	0.72 (6)	0.41 (5)	0.56 (3)	0.40 (5)	0.42 (4)	0.01	0.11	0.49
**Sweden**	3	0.48 (1)	0.52 (2)	0.66 (1)	0.79 (1)	0.48 (1)	0.57 (2)	0.47 (2)	0.45 (2)	0.01	0.12	0.55
**Denmark**	2	0.46 (2)	0.53 (1)	0.63 (2)	0.77 (2)	0.46 (2)	0.58 (1)	0.50 (1)	0.59 (1)	0.01	0.10	0.57
**Sample country average (15 countries)**	3	0.39	0.43	0.55	0.70	0.39	0.51	0.36	0.36	-	-	-

The inequality rank appears in parentheses (the most unequal country takes the value 1). Source: Authors’ elaboration based on SHARE (2021).

**Table 8 ijerph-19-07812-t008:** Inequality indicators in self-perceived health in Wave 4 (2010) for women.

Country	Women											
	m	I1	I2	I3	I4	I5-1	I5-2	I6-1	I6-2	Variance	Standard Error	Mean
**Estonia**	4	0.29 (15)	0.33 (15)	0.44 (15)	0.61 (15)	0.29 (15)	0.41 (15)	0.23 (15)	0.20 (15)	0.02	0.13	0.35
**Hungary**	4	0.42 (3)	0.43 (11)	0.58 (5)	0.73 (8)	0.42 (3)	0.52 (7)	0.41 (4)	0.39 (6)	0.01	0.12	0.49
**Poland**	4	0.38 (10)	0.39 (13)	0.51 (14)	0.65 (14)	0.38 (10)	0.48 (11)	0.35 (10)	0.32 (12)	0.01	0.11	0.43
**Portugal**	4	0.37 (12)	0.38 (14)	0.52 (13)	0.69 (13)	0.37 (12)	0.43 (14)	0.36 (8)	0.34 (10)	0.01	0.12	0.43
**Spain**	3	0.41 (5)	0.44 (9)	0.57 (7)	0.73 (8)	0.41 (5)	0.55 (4)	0.39 (5)	0.41 (4)	0.01	0.12	0.49
**Austria**	3	0.41 (5)	0.47 (3)	0.59 (4)	0.74 (6)	0.41 (5)	0.51 (10)	0.38 (7)	0.35 (8)	0.02	0.13	0.48
**Belgium**	3	0.37 (12)	0.44 (9)	0.55 (10)	0.72 (9)	0.37 (12)	0.47 (12)	0.32 (13)	0.29 (13)	0.02	0.14	0.44
**Czech Republic**	3	0.40 (7)	0.44 (9)	0.56 (9)	0.71 (11)	0.40 (7)	0.55 (4)	0.39 (5)	0.41 (4)	0.01	0.11	0.48
**France**	3	0.39 (9)	0.45 (6)	0.57 (7)	0.74 (6)	0.39 (9)	0.53 (6)	0.34 (12)	0.34 (10)	0.02	0.14	0.47
**Germany**	3	0.38 (10)	0.41 (12)	0.53 (12)	0.69 (13)	0.38 (10)	0.52 (7)	0.35 (10)	0.37 (7)	0.01	0.12	0.45
**Italy**	3	0.40 (7)	0.45 (6)	0.58 (5)	0.75 (4)	0.40 (7)	0.52 (7)	0.36 (8)	0.35 (8)	0.02	0.14	0.48
**The Netherlands**	3	0.37 (12)	0.43 (11)	0.55 (10)	0.71 (11)	0.37 (12)	0.45 (13)	0.32 (13)	0.29 (13)	0.02	0.14	0.44
**Slovenia**	3	0.42 (3)	0.46 (4)	0.60 (3)	0.76 (3)	0.42 (3)	0.57 (3)	0.41 (3)	0.43 (3)	0.02	0.12	0.51
**Sweden**	3	0.49 (1)	0.54 (1)	0.67 (1)	0.81 (1)	0.49 (1)	0.58 (1)	0.49 (2)	0.47 (2)	0.01	0.12	0.57
**Denmark**	2	0.47 (2)	0.53 (2)	0.63 (2)	0.77 (2)	0.47 (2)	0.58 (1)	0.50 (1)	0.59 (1)	0.01	0.10	0.57
**Sample country average (15 countries)**	3	0.40	0.44	0.56	0.72	0.40	0.51	0.37	0.37	-	-	-

The inequality rank appears in parentheses (the most unequal country takes the value 1). Source: Authors’ elaboration based on SHARE (2021).

**Table 9 ijerph-19-07812-t009:** Inequality indicators in self-perceived health in Wave 8 (2020) for men.

Country	Men											
	m	I1	I2	I3	I4	I5-1	I5-2	I6-1	I6-2	Variance	Standard Error	Mean
**Estonia**	4	0.28 (23)	0.32 (23)	0.43 (22)	0.60 (22)	0.28 (23)	0.38 (24)	0.22 (23)	0.19 (23)	0.02	0.13	0.34
**Latvia**	4	0.25 (25)	0.29 (25)	0.39 (25)	0.52 (25)	0.25 (25)	0.39 (23)	0.18 (25)	0.16 (25)	0.01	0.12	0.30
**Lithuania**	4	0.30 (22)	0.32 (23)	0.41 (24)	0.57 (24)	0.30 (22)	0.36 (25)	0.26 (22)	0.25 (21)	0.01	0.10	0.35
**Poland**	4	0.37 (12)	0.37 (20)	0.47 (20)	0.61 (21)	0.37 (12)	0.45 (18)	0.35 (11)	0.33 (11)	0.01	0.09	0.42
**Romania**	4	0.41 (6)	0.41 (13)	0.53 (11)	0.67 (18)	0.41 (6)	0.51 (11)	0.40 (8)	0.37 (9)	0.01	0.10	0.46
**Austria**	3	0.40 (7)	0.46 (3)	0.58 (3)	0.73 (4)	0.40 (7)	0.53 (7)	0.36 (10)	0.35 (10)	0.02	0.13	0.48
**Belgium**	3	0.35 (17)	0.42 (11)	0.53 (11)	0.69 (12)	0.35 (17)	0.47 (15)	0.30 (18)	0.28 (18)	0.02	0.14	0.42
**Bulgaria**	3	0.42 (5)	0.45 (4)	0.57 (5)	0.72 (5)	0.42 (5)	0.58 (3)	0.42 (4)	0.46 (3)	0.01	0.11	0.51
**Croatia**	3	0.44 (2)	0.44 (5)	0.57 (5)	0.71 (7)	0.44 (2)	0.59 (2)	0.46 (2)	0.53 (2)	0.01	0.10	0.52
**Cyprus**	3	0.44 (2)	0.44 (5)	0.58 (3)	0.74 (3)	0.44 (2)	0.56 (4)	0.43 (3)	0.46 (3)	0.01	0.11	0.51
**Czech Republic**	3	0.28 (23)	0.36 (21)	0.43 (22)	0.62 (20)	0.28 (23)	0.41 (22)	0.21 (24)	0.19 (23)	0.02	0.14	0.35
**France**	3	0.37 (12)	0.42 (11)	0.54 (9)	0.71 (7)	0.37 (12)	0.50 (12)	0.32 (14)	0.32 (13)	0.02	0.13	0.44
**Finland**	3	0.35 (17)	0.39 (19)	0.51 (17)	0.68 (14)	0.35 (17)	0.45 (18)	0.30 (18)	0.28 (18)	0.02	0.13	0.41
**Germany**	3	0.36 (16)	0.40 (16)	0.51 (17)	0.68 (14)	0.36 (16)	0.48 (14)	0.31 (16)	0.31 (15)	0.02	0.13	0.43
**Greece**	3	0.38 (11)	0.44 (5)	0.55 (8)	0.71 (7)	0.38 (11)	0.50 (12)	0.33 (12)	0.32 (13)	0.02	0.13	0.45
**Hungary**	3	0.40 (7)	0.41 (13)	0.53 (11)	0.68 (14)	0.40 (7)	0.55 (5)	0.41 (5)	0.45 (5)	0.01	0.10	0.48
**Italy**	3	0.40 (7)	0.40 (16)	0.52 (15)	0.69 (12)	0.40 (7)	0.52 (8)	0.39 (9)	0.42 (7)	0.01	0.10	0.47
**Luxembourg**	3	0.35 (17)	0.41 (13)	0.51 (17)	0.68 (14)	0.35 (17)	0.47 (15)	0.29 (20)	0.28 (18)	0.02	0.13	0.42
**Malta**	3	0.35 (17)	0.34 (22)	0.45 (21)	0.60 (22)	0.35 (17)	0.44 (21)	0.31 (16)	0.31 (15)	0.01	0.10	0.39
**The Netherlands**	3	0.37 (12)	0.43 (8)	0.56 (7)	0.72 (5)	0.37 (12)	0.46 (17)	0.32 (14)	0.30 (17)	0.02	0.14	0.44
**Slovakia**	3	0.34 (21)	0.43 (8)	0.53 (11)	0.71 (7)	0.34 (21)	0.45 (18)	0.28 (21)	0.25 (21)	0.02	0.15	0.42
**Slovenia**	3	0.37 (12)	0.43 (8)	0.54 (9)	0.70 (11)	0.37 (12)	0.52 (8)	0.33 (12)	0.33 (11)	0.01	0.13	0.45
**Spain**	3	0.40 (7)	0.40 (16)	0.52 (15)	0.67 (18)	0.40 (7)	0.55 (5)	0.41 (5)	0.45 (5)	0.02	0.10	0.48
**Sweden**	3	0.43 (4)	0.49 (2)	0.62 (2)	0.77 (1)	0.43 (4)	0.52 (8)	0.41 (5)	0.39 (8)	0.01	0.13	0.51
**Denmark**	2	0.49 (1)	0.52 (1)	0.63 (1)	0.77 (1)	0.49 (1)	0.60 (1)	0.55 (1)	0.69 (1)	0.01	0.10	0.59
**Sample country average (25 countries)**	3	0.37	0.41	0.52	0.68	0.37	0.49	0.34	0.35	-	-	-

The inequality rank appears in parentheses (the most unequal country takes the value 1). Source: Authors’ elaboration based on SHARE (2021).

**Table 10 ijerph-19-07812-t010:** Inequality indicators in self-perceived health in Wave 8 (2020) for women.

Country	Women											
	m	I1	I2	I3	I4	I5-1	I5-2	I6-1	I6-2	Variance	Standard Error	Mean
**Estonia**	4	0.27 (24)	0.31 (24)	0.42 (23)	0.59 (22)	0.27 (24)	0.39 (23)	0.20 (24)	0.18 (24)	0.02	0.13	0.33
**Latvia**	4	0.25 (25)	0.29 (25)	0.37 (25)	0.49 (25)	0.25 (25)	0.35 (24)	0.18 (25)	0.15 (25)	0.01	0.11	0.29
**Lithuania**	4	0.30 (22)	0.32 (23)	0.42 (23)	0.59 (22)	0.30 (22)	0.35 (24)	0.28 (20)	0.26 (20)	0.01	0.11	0.35
**Poland**	4	0.36 (14)	0.37 (21)	0.46 (21)	0.60 (21)	0.36 (14)	0.45 (18)	0.34 (8)	0.32 (12)	0.01	0.09	0.41
**Romania**	3	0.40 (5)	0.43 (8)	0.55 (8)	0.69 (12)	0.40 (5)	0.57 (1)	0.39 (4)	0.43 (3)	0.01	0.11	0.48
**Austria**	3	0.39 (7)	0.45 (5)	0.57 (5)	0.73 (5)	0.39 (7)	0.51 (7)	0.34 (8)	0.33 (9)	0.02	0.14	0.46
**Belgium**	3	0.33 (20)	0.40 (16)	0.50 (17)	0.67 (16)	0.33 (20)	0.43 (21)	0.26 (22)	0.24 (21)	0.02	0.14	0.40
**Bulgaria**	3	0.40 (5)	0.46 (4)	0.57 (5)	0.72 (7)	0.40 (5)	0.54 (4)	0.37 (6)	0.37 (5)	0.02	0.12	0.48
**Croatia**	3	0.42 (3)	0.43 (8)	0.57 (5)	0.73 (5)	0.42 (3)	0.56 (3)	0.42 (2)	0.46 (2)	0.01	0.11	0.50
**Cyprus**	3	0.44 (2)	0.49 (2)	0.63 (1)	0.78 (1)	0.44 (2)	0.56 (2)	0.41 (3)	0.40 (4)	0.02	0.13	0.52
**Czech Republic**	3	0.29 (23)	0.38 (19)	0.45 (22)	0.62 (20)	0.29 (23)	0.44 (19)	0.23 (23)	0.21 (23)	0.02	0.14	0.36
**France**	3	0.35 (15)	0.42 (11)	0.53 (11)	0.70 (10)	0.35 (15)	0.49 (12)	0.30 (17)	0.29 (17)	0.02	0.14	0.43
**Finland**	3	0.35 (15)	0.38 (19)	0.49 (18)	0.67 (16)	0.35 (15)	0.47 (16)	0.31 (16)	0.30 (15)	0.02	0.12	0.42
**Germany**	3	0.38 (9)	0.41 (12)	0.53 (11)	0.69 (12)	0.38 (9)	0.51 (7)	0.35 (7)	0.35 (7)	0.01	0.12	0.45
**Greece**	3	0.37 (11)	0.44 (7)	0.55 (8)	0.71 (8)	0.37 (11)	0.49 (12)	0.32 (14)	0.30 (15)	0.02	0.14	0.44
**Hungary**	3	0.35 (15)	0.39 (18)	0.49 (18)	0.66 (18)	0.35 (15)	0.49 (12)	0.32 (14)	0.32 (12)	0.01	0.12	0.42
**Italy**	3	0.38 (9)	0.40 (16)	0.53 (11)	0.70 (10)	0.38 (9)	0.50 (9)	0.34 (8)	0.35 (7)	0.02	0.12	0.45
**Luxembourg**	3	0.35 (15)	0.41 (12)	0.53 (11)	0.59 (22)	0.35 (15)	0.47 (16)	0.30 (17)	0.28 (18)	0.01	0.11	0.41
**Malta**	3	0.35 (15)	0.37 (21)	0.49 (18)	0.64 (19)	0.35 (15)	0.44 (19)	0.29 (19)	0.28 (18)	0.01	0.12	0.40
**The Netherlands**	3	0.39 (7)	0.45 (5)	0.58 (4)	0.74 (4)	0.39 (7)	0.48 (15)	0.34 (8)	0.32 (12)	0.02	0.14	0.46
**Slovakia**	3	0.33 (20)	0.41 (12)	0.51 (16)	0.69 (12)	0.33 (20)	0.41 (22)	0.27 (21)	0.24 (21)	0.02	0.15	0.40
**Slovenia**	3	0.37 (11)	0.43 (8)	0.54 (10)	0.71 (8)	0.37 (11)	0.52 (6)	0.33 (12)	0.33 (9)	0.02	0.13	0.45
**Spain**	3	0.37 (11)	0.41 (12)	0.52 (15)	0.69 (12)	0.37 (11)	0.50 (9)	0.33 (12)	0.33 (9)	0.02	0.12	0.44
**Sweden**	3	0.41 (4)	0.47 (3)	0.60 (3)	0.75 (3)	0.41 (4)	0.50 (9)	0.38 (5)	0.36 (6)	0.02	0.13	0.49
**Denmark**	3	0.48 (1)	0.52 (1)	0.63 (1)	0.77 (2)	0.48 (1)	0.54 (4)	0.50 (1)	0.47 (1)	0.01	0.10	0.55
**Sample country average (25 countries)**	3	0.36	0.41	0.52	0.68	0.36	0.48	0.32	0.31	-	-	-

The inequality rank appears in parentheses (the most unequal country takes the value 1). Source: Authors’ elaboration based on SHARE (2021).

## Data Availability

The data used can be obtained from the website of http://www.share-project.org/home0.html (accessed on 22 June 2022).

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
