# Peer review of "Have Health Inequalities Increased during the COVID-19 Pandemic? Evidence from Recent Years for Older European Union Citizens"

_ijerph, 2022, doi:10.3390/ijerph19137812_

Round 1

Reviewer 1 Report

1. Some recent literature of WHO and Euro Health Net could be mentioned

2. The inequality indices need explanation

3. Methods need more explanation. Are there secular trends in self perceived health? How are ages standardized in the different years? What kind of corrections are made for the different countries in the samples of the different years? 

4. Some tables (e.g. 2-5) seem redundant, or could better be placed in an appendix.

5. In the conclusions the authors state that inequalities deminished as a result from governmental policy. But this is more an assumption than a conclusion, since they provide no evidence about causal influences. External factors might also have had a lowering influence. This paper is not about effects, but only provide some descriptive evidence.

Author Response

Dear Editor,

We are truly thankful for your positive reply. Let us explain below how we have decided to address the comments and suggestions of the Editor and the reviewer.

  1. Some recent literature of WHO and Euro Health Net could be mentioned

As proposed by the editor, we have inserted some new ideas. Both the World Health Organisation and Euro Health Net have been discussing in recent months how we could improve the situation of equity and social welfare, always in terms of health. In this sense, and in response to your recommendation, we have included a few lines in the conclusion citing the main ideas in this regard:

“In this regard, it is worth noting that in the wake of the COVID-19 pandemic, major international organisations are discussing how public health and social determinants could be improved across Europe. This is the case of EuroHealthNet, which is looking at how countries could come together to create enabling environments for wellbeing, with equity and wellbeing as key pillars for the development and promotion of such public policies”.

  1. The inequality indices need explanation

A large literature review has been carried out to find indices that measure inequality and that can be applied, firstly, to the field of health and, secondly, that accept ordinal variables. This second aspect was more complicated and limited the analysis to a greater extent. Finally, we opted for the indices that appear in the article because of the use that has been made of them by other leading authors. All the citations mentioned in the text allow the reader to research and read more information about them. Some of them are:

Apouey, B. (2007). Measuring health polarization with self-assessed health data. Health Econ. 16, 875–894.

Naga, R. H. A., & Yalcin, T. (2008). Inequality measurement for ordered response health data. Journal of Health Economics, 27(6), 1614-1625.

Reardon, S.F. (2009). Measures of ordinal segregation. Res Econ. Inequal. 17, 129–155.

Lazar, A., Silber, J. (2013). On the cardinal measurement of health inequality when only ordinal information is available on individual health status. Health Econ. 22, 106–113.

  1. Methods need more explanation. Are there secular trends in self perceived health? How are ages standardized in the different years? What kind of corrections are made for the different countries in the samples of the different years? 

The SHARE waves not included in this analysis corroborate the trend presented in this article. In general terms, we could note that men tend to self-report better health than women. In addition, having a low level of education or income tends to be associated with self-reported poor health. These ideas are also shown in data from other sources, e.g. EUROSTAT. (https://ec.europa.eu/eurostat/statistics-explained/index.php?title=File:Self-perceived_health_2020.png) or Ph, P. D. E. K. D., & Dimitrova, D. E. K. (2021). Social differences in self-perceived health of people at older age in European societies. Results of the survey of health, ageing and retirement in europe (share). Revista Inclusiones, 01-21.

The literature on these issues could be divided into those that standardise ages across years and those that do not. In view of the graphical analysis we have done, this second option was the best way to approach it. All individuals in the sample are over the age of fifty in the three selected waves (2006, 2010 and 2020) in order to focus on a more vulnerable group of the population than the young.

After extracting the data from the SHARE Survey database, they have been preprocessed for the correct construction of this analysis. In a first phase, all individuals with incomplete responses (empty cells) have been eliminated. In a second phase, those individuals with invalid responses for our analysis (i.e. those under 50 years of age) have been eliminated. Ultimately, individuals from Israel have been eliminated, as in our sample we are limited to European countries to focus our analysis on this geographical area. Again, this is something that is done in other similar papers, as in the case of: Quality of Life, Health and the Great Recession in Spain: Why Older People Matter? Carla Blázquez-Fernández David Cantarero-Prieto Marta Pascual-Sáez  Int. J. Environ. Res. Public Health 2021, 18(4), 2028; https://doi.org/10.3390/ijerph18042028 (www.mdpi.com/1660-4601/18/4/2028)

  1. Some tables (e.g. 2-5) seem redundant, or could better be placed in an appendix.

Tables 2, 3 and 4 have been moved to the annex. Table 5 has been left in the text as it shows the results of the indices for the first wave studied.

  1. In the conclusions the authors state that inequalities deminished as a result from governmental policy. But this is more an assumption than a conclusion, since they provide no evidence about causal influences. External factors might also have had a lowering influence. This paper is not about effects, but only provide some descriptive evidence.

According to what we have been suggested, the statement is replaced by a possible future line assessing the effects of specific public policies on health inequalities.

Reviewer 2 Report

Have health inequalities increased after the COVID-19 pandemic?  Evidence from recent Years for Older European Union Citizens”, Submitted to the International Journal of Environmental Research and Public Health

The analysis computes six indices of health inequality subject to the indices having desirable theoretical properties (lines 164 to 167).  All indices are functions of the proportions of the population in five categories of self-assessed health, excellent, very good, good, fair, and poor.  The authors correctly state that self-assessed health is valid and predicts future morbidity and mortality.  However, there are no citations.  I agree completely with the statements made but I want citations to literature.

The paper is generally excellent but has no standard errors of the estimates.  Here is how to compute the standard errors.

The input data are the proportions of the population in the five categories in various European countries in each of three years, 2002, 2010, and 2020.  Call those five proportions pj, 1 ≤ j ≤ 5.  They are estimating population parameters πj, 1 ≤ j ≤ 5.  That is a multinomial distribution.  

Let the sample size in a particular country be n.  The asymptotic distribution of the estimates is

V(pj) = pj(1 – pj)/n, like the standard Binomial result in statistics, and

Cov(pj, pk) = − pjpk.

The variances are like Binomials, and the covariances are negative because all five probabilities must add up to 1.0, so an increase in one probability must reduce some other probability. 

The multinomial distribution can be found many places, but one of my favorite sources is Bishop, Feinberg, and Holland, Discrete Multivariate Analysis:  Theory and Practice (1975), pages 469-472.  Kendall’s Advanced Theory of Statistics (1994), 6th edition, Volume 1, has section 7.7, page 260.  Wikipedia’s article on Multinomial Distribution is very straightforward..

The point is that the p’s have a probability distribution, and the six measures, inequality measures (1) to (6) in the paper are functions of the p’s, so that implies variances of the inequality measures.  The variances are given by the delta method, as the inequality measures are non-linear functions of the p’s.  The delta method is also in Bishop, Feinberg, and Holland, and in Wikipedia, and both use the Binomial as an example.  The multinomial is just a Binomial with more than two categories.

So compute the variances and standard errors (square roots of the variance) for countries and measures, and this paper will be as good statistically as it is now in data quality.

The writing

The writing is excellent and highly readable, but there is an editing error around lines 185 to 190 with duplicated lines.

Tables 5 to 7 are difficult to read because too much is presented to put the numbers on one line.  One solution is to put the data on pages in landscape orientation.  If that does not work, just put the rank on a second line in all cases, like Spain in Table 6.

Figures 3 to 5 are mislabeled in some way.  The heading says “indicator for women” but the figure states that it shows both men and women.  The heading appears to be incorrect.

Reviewer 3 Report

This is an interesting paper about health inequality metrics based on ordinal self-assessed health data from a range of European countries, finding an apparent trend for reduced health inequalities in the 2006-2020 period. 

I suggest a few modifications of the paper as detailed below.

1. The last included SHARE wave was captured in 2020, therefore all mentions of  "after the COVID-19 pandemic" in the title and text shall be changed to "during the COVID-19 pandemic", or "in the first year of COVID-19 pandemic", etc.)

2. The analysis is focused on within-country inequality indices and their trends, separately for men and women, without considering inequalities between countries. As the Authors pointed out, Šucur and Zrinšcak [7] warned that results were worse in Croatia than in the European comparator countries; and Pinillos-Franco and Somarriba-Arechavala described a territorial pattern across Europe (Eastern countries had poorer levels of health). These previous findings could have some reactions in the analyses and/or in the discussion, or at least a corresponding statement on the limitations of the presented analyses. 

3.  In line 165, all the indices are stated to satisfy simple aversion to median-preserving spreads. This feature shall be explained better - at the first glance, being insensitive to changes in distributions when the median is unchanged is not an advantage for indices that aim to measure inequity (~ spread of the distribution). 

4. It would be helpful to add average index columns (by gender) to Tables 6-7. 

5. Line 247: "the growth rates of inequalities in SAH are not positive in 2006, 2010 and 2020." - the changes were analysed in the 2006-2020 period, so growth rate in 2006 was not investigated - suggest changing to "not positive in the 2006-2020 period". 

6. Lines 254-270: please clarify in the text what "cases" means in this paragraph. It may be misunderstood by the reader - associating e.g., to survey respondents in the various SHARE waves. 

7. Figures 10-11: country names are missing for every second country. 

Round 2

Reviewer 2 Report

Everything is improved except that I cannot find any standard errors in Tables 5 to 7.  The authors said they computed standard errors:  "As suggested by the editor, variances and standard errors have been added in tables 5-10...."  Where are Tables 8 to 10?

In my original report, I left out a division by n.

Cov(pj, pk) = − pjpk/n.  All variances and covariances go to zero as n goes to infinity, i.e. if n is very large, every category is asymptotically independent of the other categories.  Standard errors still need to be computed.

Author Response

Dear Editor, 

Thank you very much for your comments and your help. In the first revision, there was an error in modifying the tables. Please find attached the version with all tables 5-10. As suggested by the reviewer, the variance and standard errors have been incorporated. As suggested by another reviewer, country averages have also been added.
